# LLM AS A COMPLEMENTARY OPTIMIZER TO GRADIENT DESCENT: A CASE STUDY IN PROMPT TUNING

## ABSTRACT

Mastering a skill generally relies on both hands-on experience from doers and insightful, high-level guidance by mentors. *Will this strategy also work well for solving complex non-convex optimization problems?* Here, a common gradient-based optimizer acts like a disciplined doer, making locally optimal updates at each step. Large Language Models (LLMs) can also search for better solutions by inferring from natural language instructions, akin to a high-level mentor. In this paper, we show that these two participators are complementary to each other and can effectively collaborate as a combined optimization framework. The collaborative optimization is achieved by alternating between the gradient-based and LLM-based optimizers. We instruct LLMs to generate possibly improved solutions by taking parameter trajectories recorded during the previous stage of gradient-based optimization into account. Inferred results of LLMs are used as restarting points for the next stage of gradient optimization. We verify the effectiveness of this optimization framework on prompt tuning. By leveraging both the locally rigorous gradient-based optimizer and the high-level deductive LLM-based optimizer, the combined optimization method consistently yields improvements over competitive baselines on a variety of tasks. Our results demonstrate the synergistic effect of conventional gradient-based optimization and the inference ability of LLMs.

The code will be made publicly available.

## 1 INTRODUCTION

Humans acquire skills through practical experience and external guidance from mentors. Similarly, solving optimization problems relies on well-designed algorithms incorporating prior knowledge, as well as meticulous procedural implementation. Practically, gradient-based algorithms have almost become the default choice for solving optimization problems in various machine learning models. We regard the gradient-based optimizers as disciplined doers that are effective in navigating the parameter space through precise, incremental adjustments based on gradient information. However, their local perspective often limits their ability to escape local optima and discover more optimal solutions.

In this work, we proposed an optimization method using LLMs as optimization instructors to provide high-level guidance for gradient-based optimizers. The basis of LLMs capable of solving optimization problems lies in their ability to comprehend and generate nuanced and contextually relevant text. Recent studies have proposed to utilize LLMs as strategy planners or optimizers in concrete optimization tasks. For example, Eureka (Ma et al., 2024) trains agents by reinforcement reward function designed by GPT-4, which can learn complex skills such as dexterous pen spinning. It shows that LLM can guide the trend of optimized policy on a delicate level.

Employing LLMs for optimization offers unique advantages. The optimization is conducted with natural language interactions, which contributes to two charming properties. First, the implementation of the optimization is code-free. The optimization process only involves natural language instruction-response interactions with LLMs. Secondly, LLMs generate instruction-related outputs by assembling task-related semantic tokens that are difficult to discover through continuous gradient-based learning. The generation results can be diverse and hardly limited by the local optima issue, which is often encountered by gradient-based optimization. The solutions discovered by LLM with a lower loss value have more possibilities to optimize to a better convergence point.

On the other hand, LLM-based optimization faces instability issues since it has no lexical constraints. LLMs analyze the problem on the semantic level and response in vocabulary space. The generated

solutions may not be as precise as the results optimized by rigorous step-by-step gradient descent in the parameter space, especially under limited LLM API calling budgets. Existing LLM-based methods, such as Liu et al. (2023a), need to set multiple search trials to find a promising result. We instruct LLMs with the optimization trajectory of gradient-based optimizers, which is converted to natural language format, grounding LLMs to a more promising sub-region of the vocabulary space. The local carefulness of gradient-based optimizer and diverse semantic exploration of LLM-based optimizer are complementary to each other, suggesting a collaborative optimization approach.

With this motivation, we propose an optimization method that combines the conventional gradient-based optimizer and LLM optimizer. The optimization approach leverages both the locally rigorous gradient-based optimizer in parameter space and high-level deductive LLM-based optimizer in unconstrained vocabulary space for better performance. To achieve collaborative training based on the two optimizers, we interleave the conventional training process of gradient-based optimization with interactions with LLM. First, we optimize the parameters for only dozens of iterations using a gradient optimizer. Then the optimized parameters in the intermediate step, along with their loss and accuracy on the training set, are provided as history trajectory clues for LLM to infer new candidates that are potentially more effective. After grabbing the response from LLM, we use the generated results as restarting points of the parameters for subsequent gradient-based optimization iterations. The two optimizers are operated alternately to optimize the parameter collaboratively. The final optimized results are obtained by the gradient optimizer with a stable convergence. Our proposed optimizing strategy only injects several API calls to LLM to the conventional gradient-based training workflow.

We study the effectiveness of the combined optimization in a widely considered prompt tuning framework. Specifically, we optimize textual prompts for language and vision-language pre-trained models. Tuning such prompts is shown to bring significant performance improvements for the adaptation of pre-trained models (Zhou et al., 2022b; Lester et al., 2021). However, optimizing the prompt in the input discrete vocabulary space or word embedding space is not an easy problem for conventional gradient-based optimizers that is widely adopted. We validate that the proposed collaborative optimization framework leads to consistent improvements in prompt optimization across a variety of tasks and optimizer LLMs.

In summary, our contributions include:

- Based on the textual prompt optimization problem, we showcase the limitations of gradient-based optimization, especially the entrapment in local optima, and attribute the issues to the limitation of gradient-based optimizer in the short-sighted local perspective of the parameter space.

- We propose a novel optimization approach that combines the deductive LLM-based optimizer in unconstrained vocabulary space with the disciplined gradient-based optimizer in the parameter space for better optimization performance.

- We test the effectiveness of the proposed combined optimization method on prompt tuning tasks, and it achieves consistent improvements over existing competitive baseline methods, validating the complementary effect of LLM-based and gradient-based optimizers.

## 2 RELATED WORK

### 2.1 LLMS AND OPTIMIZATION PROBLEMS

Recent developments of Large Language Models (LLMs) have demonstrated an unprecedented ability to comprehend and generate human-like text, leading to significant breakthroughs in natural language processing (Touvron et al., 2023a; Chowdhery et al., 2022). The robust capability of LLMs in natural language comprehension and the generation of more nuanced and contextually relevant text provides a foundation for various advanced open-ended applications, where they are being instructed to participate in dialogue (OpenAI et al., 2024), formulate and execute plans (Gupta & Kembhavi, 2022; Gao et al., 2023), writing codes (Ma et al., 2024), etc. LLMs' rich prior knowledge and reasoning ability open the way to addressing practical optimization problems for real-world applications. Existing works have validated the effectiveness of LLM for solving small-scale mathematical optimization problems (Yang et al., 2023), optimizing prompts (Zhou et al., 2023; Pryzant et al., 2023; Guo et al., 2024; Liu et al., 2023a; Fernando et al., 2023; Diao et al.,

2023), searching for network architectures (Chen et al., 2023; Zheng et al., 2023), hyperparameter optimization (Chen et al., 2022) and discovering physical equations (Du et al., 2024).

In terms of prompt optimization, APE (Zhou et al., 2023) proposes to use LLMs to generate and select natural language prompts by instructing LLMs with task definitions and targets. LLMs can obtain better solutions iteratively by analyzing previously found candidates. APO (Pryzant et al., 2023) proposes that editing prompts by LLM is analogous to conducting gradient descent in the natural language domain. They imitate the gradient-based learning by providing the failure cases to LLM for a semantic "gradient" and updating the prompt in an opposite semantic direction. EVOPROMPT (Guo et al., 2024) also connects LLM-based optimization to traditional algorithms for better explainability. They integrate LLM into the workflow of evolutionary algorithms by instructing LLM to act like evolutionary operators to generate new candidate prompts. The insight that LLM naturally enables an intelligent variation operator is also revealed in LMC (Meyerson et al., 2024) and ELM (Lehman et al., 2022) on image and code generation tasks. Liu et al. (2023a) searches prompts for the vision-language model by conversing with LLM following designed strategies and achieves comparable results to white-box gradient-based prompt tuning.

The results achieved in these approaches demonstrate that LLMs can be applied as a general-purpose optimizer for optimization tasks. Although some of them (Pryzant et al., 2023; Guo et al., 2024) explored the connection between LLM-based inference and conventional optimization algorithms, e.g., gradient descent, evolutionary computing. However, the proposed optimization workflows are still largely based on the inherent ability of LLM, which leads to inadequate data utilization and suboptimal performance. For example, in Liu et al. (2023a), the performance superiority only holds in the one-shot training set and LLMs can not effectively optimize to gain more improvements based on more training data. It shows that the interaction format of natural language makes it hard for LLMs to optimize as precisely as numerical optimization algorithms, e.g., gradient-based optimizers. Besides, the API calling budget bounded by the high cost of operating super large-scale models also limits the performance of LLM-based optimization. This motivates us to design a collaborative optimization method to achieve better optimization performance by combining both the results of LLMs' high-level reasoning and the stable convergence of conventional gradient-based optimizers.

## 2.2 Prompt Tuning for Pre-trained Models

Prompt tuning has emerged as a standard approach for the parameter-efficient adaptation of pre-trained models, aimed at improving their performance in various natural language processing (Lester et al., 2021; Li & Liang, 2021) and vision-language (Zhou et al., 2022b;a; Yao et al., 2024) tasks. Prompt-based tuning of pre-trained models appends learnable embeddings to the original sequence of the data for the input layer or intermediate layer. Fine-tuning the lightweight parameters in the prompt yields comparable performance even to full parameter fine-tuning and transferability (Vu et al., 2022; Su et al., 2022) on various tasks (Lester et al., 2021; Li & Liang, 2021; Liu et al., 2022). Despite its widespread adoption, the conventional prompt tuning technique encounters challenges related to slow convergence and suboptimal optimization (Ding et al., 2022), which undermines the effectiveness of prompt tuning in a wider and larger scale of pre-trained models and downstream tasks. We attribute these issues to the complexity of the input embedding space of the pre-trained model, making it challenging to optimize the prompt effectively based on back-propagated gradients in this space.

## 3 Method

In this section, we introduce our proposed combined optimization approach that leverages both the local carefulness of gradient-based optimizer and the flexible semantics exploration of LLM-based optimizer. The overview of our method is shown in Figure 1. We instantiate the problem in a prompt tuning scenario to elaborate on our proposed method. We will describe the general formulation of prompt tuning/optimization and the way of gradient-based prompt tuning in Section 3.1. Next, we analyze the issues that occur in the conventional gradient-based prompt tuning process in Section 3.2, and attribute the problem to the characteristics of gradient-based optimizer that is limited to the local view of the parameter space. Finally, we introduce our proposed combined optimization method in Section 3.3.

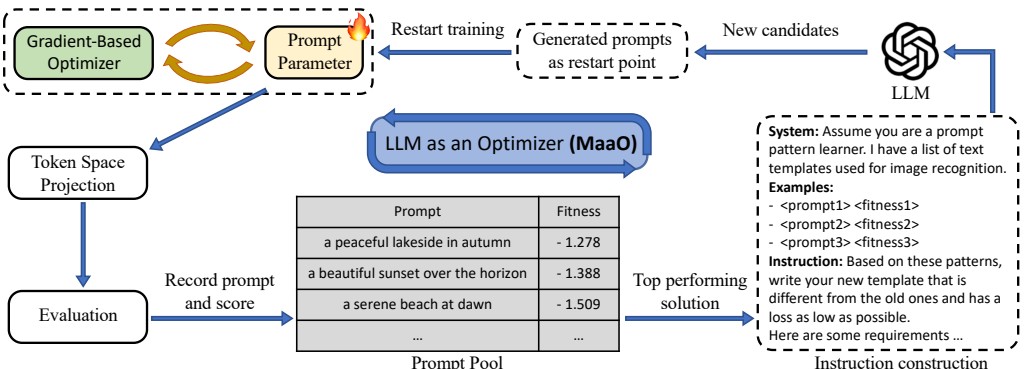

Figure 1: Overview of our proposed method. The bold arrows with different color show the two collaborative optimizers of in our method. The thin arrows show the workflow of MaaO which infer for promising candidate prompt in vocabulary space for gradient-based optimizer.

### 3.1 GENERAL FORMULATION OF PROMPT TUNING

In this part, we instantiate the task by prompt tuning for discriminative tasks, i.e., classification. In a general situation, we consider a pre-trained multi-modal model $\mathcal{E}$. The classes of input images $I$ or texts $T$ can be recognized by classifying the representations $\mathcal{E}(I, T)$, encoded by the pre-trained model. We denote the task-specific classifier as $\mathcal{F}(\cdot)$. The prediction can be obtained by $p(\hat{y}|I, T) = \mathcal{F}(\mathcal{E}(I, T))$.

To better adapt pre-trained models to various downstream tasks, prompt tuning introduces learnable prompt tokens and formulates a task-specific input for the pre-trained model. The learnable prompt tokens can be either continuous vectors (Zhou et al., 2022b; Lester et al., 2021) in the textual embedding space of the pre-trained model or discrete tokens (Diao et al., 2022; Deng et al., 2022) sampled from the vocabulary. The prompt $P$ parameterized by $\boldsymbol{\theta}$ is concatenated with the original input making up a task-specific input. The adapted output can be formulated as $p(\hat{y}|I, T; \boldsymbol{\theta}) = \mathcal{F}(\mathcal{E}(P_{\boldsymbol{\theta}}, I, T))$. In common practice, the prompt tokens are learned through labeled few-shot samples from target task datasets. The parameters of the prompt are optimized by minimizing the loss function:

$$\boldsymbol{\theta}^* = \arg\min_{\boldsymbol{\theta}} \mathcal{L}(y, I, T, \boldsymbol{\theta}) = \arg\min_{\boldsymbol{\theta}} -\log p(\hat{y} = y|I, T; \boldsymbol{\theta}). \tag{1}$$

According to this formulation, it is straightforward to use a standard gradient-based optimizer to learn the parameters as is done in conventional prompt tuning methods:

$$\boldsymbol{\theta}_{t+1} = \boldsymbol{\theta}_t - \eta_t \nabla_{\boldsymbol{\theta}} \mathcal{L}(y, I, T, \boldsymbol{\theta}). \tag{2}$$

### 3.2 ANALYSIS ON ISSUES OF GRADIENT-BASED PROMPT TUNING

Although prompt tuning has become one of the most widely adopted parameter-efficient fine-tuning methods for the adaptation of pre-trained models. The optimization of the prompt still encounters challenges. The prompts converge much slower than other parameters efficient fine-tuning methods, e.g., adapter tuning or even full parameter fine-tuning (Ding et al., 2022), based on the estimated gradients back-

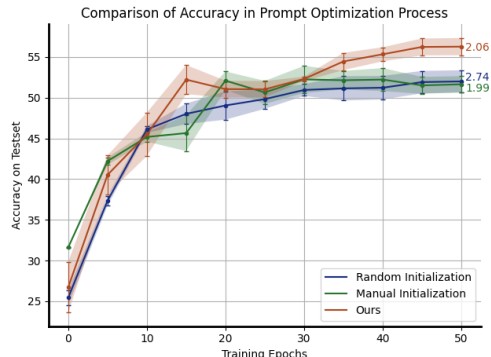

Figure 2: The result of gradient-based prompt optimization with different prompt initialization. The shadow denotes the standard deviation of the accuracy over three random seeds.

propagated through the entire pre-trained model. Another main issue of prompt tuning is that the effectiveness of the learned prompt is sensitive to its initialization values, suggesting that the optimization of the prompt may easily entrapped in local optima due to the complexity of the embedding space of the pre-trained model. Unfortunately, it is challenging to carefully craft initial prompts for every

Hi GPT, assume you are a prompt pattern learner. I have a list of text templates with their corresponding loss values and accuracy. They are used for image classification with CLIP model. The templates are arranged in descending order based on their loss value on training samples, where lower loss indicates better quality.

Templates: a precise satellite view of
Loss: 2.18
Accuracy: 20.0

Templates: a centered satellite photo of {}. (Mamual prompt to inject prior knowledge.)
Loss: 1.96
Accuracy: 30.0

Templates: a crisp high - definition image of
Loss: 1.85
Accuracy: 50.0

... (more optimized prompts and scores)
There are latent patterns that make the template good. Based on these patterns, write your new template that is different from the old ones and has a loss as low as possible.
Here are some requirements
- Please reply with only the template
- Keep every template under 10 words
- Generate 3 templates that potentially have better image classification performance

Figure 3: The instruction used to query GPT-3.5 and GPT-4.0 in an iteration of optimizing the prompt using LLM.

downstream task. To address this issue, Gu et al. (2022) propose to seek a satisfying initialization point for the prompt. However, their method needs to inject soft prompts into the pre-training stage, which limits its application to scenarios where pre-training resources are limited.

To demonstrate the issues more specifically, we analyze some empirical results of gradient-based prompt optimization performed on the one-shot training set of EuroSAT (Helber et al., 2019). We fix the training set for all experiments to eliminate the variance caused by data sampling. We run CoOp (Zhou et al., 2022b) under three random initializations and show the results as indicated by "Random Initialization" in Figure 2. It can be seen that even if we fix the training samples, different random initialization values of the prompt can still bring considerable standard deviation in the results of final learned prompts, indicating a large performance gap (up to 9 percent of accuracy) between different seeds. If we manually initialize the prompt as a prompt template "a photo of a", which is used in Radford et al. (2021)'s work, the final variance gets smaller but the absolute performance shows a slight decline. Prior knowledge contained in manual prompts brings merits, providing better results at the starting phase of the training, but lacks proper flexibility for enhancement of final learned prompts. Our method adds marginal steps of optimization based on the collaboration of gradient-based optimizer and MaaO at the start of the training workflow, which results in both lower standard deviation and better absolute performance.

The high sensitivity of prompt tuning results according to different initialization values indicates the complexity of the input embedding space, where gradient-based optimizer only leads to suboptimal converged parameters based on gradient information in a short-sighted local perspective, hardly considering the semantics of the prompt and the overall task information. To mitigate the limitations of the gradient-based optimizer, we leverage LLM as an unconstrained vocabulary space prompt optimizer based on textual semantic information of the task and previously found prompts.

### 3.3 A Collaborative Optimization Method by Using LLM as a Prompt Optimizer

We propose to harness LLM as an optimizer (MaaO) to mitigate the issues of gradient-based prompt tuning. We leverage the unconstrained inductive ability of LLM in vocabulary space based on high-level semantic information of the prompt to complement the gradient-based optimizer.

Our method optimizes the prompt by using the gradient optimizer and MaaO in an alternating pattern. Specifically, we first update the parameter of the prompt for minor steps of gradient-descent optimization and record the intermediate learned prompts and corresponding fitness scores, which are evaluated on the few-shot training samples. Then, we construct instruction for LLM with the intermediate learned prompts as optimizing trajectory information. Taking the instruction as input, LLM generates more promising candidate prompts for the target model. Next, we reinitialize the parameter of the prompts with LLM-generated prompts and restart the gradient-based training process for the next round. After operating the above two optimizers alternately for few rounds, we finally train the prompt to convergence using the gradient-based optimizer. In the following, we will describe the components of MaaO and show their combination with the gradient-based optimizer for collaborative prompt optimization.

**Instruction construction.** The gradient optimizer calculates updates based on the current parameters and objective function. Information on the current state of optimization should also be properly provided for LLM to infer from. We collect the intermediate optimized prompt in the training trajectory of the gradient-based optimizer and evaluate the performance corresponding to each intermediate prompt as a fitness score, indicating how good or bad the prompt performs. Considering that the accuracy may not be precise enough on a few samples, we employ loss as the indicator value. LLM is instructed to generate prompts that potentially achieve better performance based on observed patterns in top-performing candidates.

We also briefly define the role of LLM and explain the optimization goal in natural language, encouraging LLM to assemble task-related tokens when constructing the prompt. Additional instructions to constrain the length and number of the generated prompts are included for programmed processing. Figure 3 shows the instruction used in each optimization iteration of the prompt using GPT-3.5 and GPT-4.

---

**Algorithm 1** Combined Optimization Algorithm

**Require:** Prompt $p_\theta$ parameterized with $\theta$, training set $\mathcal{D}$, loss function regarding the target pre-trained model and the training set $f_\mathcal{D}(\cdot)$, number of optimization rounds $N$, number of iterations $m$, $M$, embedding layer operater $\mathcal{V}(\cdot)$ and token space projection operator $\mathcal{V}^{-1}(\cdot)$, prompt candidates set $\mathcal{P}$.

1: **Initialize:** prompt $\theta$ with random values, $\mathcal{P} \leftarrow \varnothing$.
2: **for** $n = 1$ to $N$ **do**
3:     **// Gradient-based optimization:**
4:     **for** $\tau = 1$ to $m$ **do**
5:         Update: $\theta_\tau \leftarrow \theta_{\tau-1} - lr \cdot \nabla f(\theta_{\tau-1})$
6:         Record: $\mathcal{P} \leftarrow \mathcal{P} \cup \{(p_\theta, -, -)\}$
7:     **// Prompt evaluation:**
8:     **for** $p_\theta$ in $\mathcal{P}$ **do**
9:         Discretize: $\hat{p} \leftarrow \mathcal{V}^{-1}(p_\theta)$
10:       Evaluate: $s \leftarrow f_\mathcal{D}(\hat{p})$
11:       Record: $\mathcal{P} \leftarrow \mathcal{P} \cup \{(p_\theta, \hat{p}, s)\}$
12:     **// LLM-based optimization:**
13:     Sample: $\{\hat{p}_i\}_{i=1}^k \leftarrow \text{TopK}_s(\hat{p}|(p_\theta, \hat{p}, s) \in \mathcal{P})$
14:     Generate: $\widetilde{p} \leftarrow \text{LLM}(\text{Instruction}(\{\hat{p}_i\}_{i=1}^k))$
15:     Reinitialize: $p_\theta \leftarrow \mathcal{V}(\widetilde{p}), \mathcal{P} \leftarrow \varnothing$
16: **// Gradient-based optimization:**
17: Train the prompt parameter with gradient optimizer for $M$ iterations till convergence.
18: **Return** the optimized prompt $p_\theta^*$

---

**Token space projection.** Gradient-based prompt tuning typically optimizes continuous prompt embeddings in the token space of the pre-trained model. However, it is not feasible to directly provide the soft embedding vectors as input to the LLM, which is proficient in responding to natural language with semantics. To convert the soft prompt embedding to discrete words, we employ a reverse process of word2vec (Mikolov et al., 2013) to project the embedding to the matched vocabulary.

Given a pre-trained target model with token embedding layer $\mathcal{V}(\cdot)$, textual inputs $\{t_i\}_{i=1}^l$ to the model are first converted to vector sequence as $\{t_i\}_{i=1}^l = \{\mathcal{V}(t_i)|i \in [1, l]\}$, before input into the model. Gradient-based prompt tuning optimizes in the continuous vector space for best prompt embeddings. We define an inverse projection function $\mathcal{V}^{-1}(\cdot)$ to project the continuous prompt vector $\hat{t}_i$ to nearest discrete tokens by $\hat{t}_i = \mathcal{V}^{-1}(\hat{t}_i)$. $\mathcal{V}^{-1}$ is defined as:

$$\mathcal{V}^{-1}(\hat{t}) := \underset{\hat{t} \in \mathcal{S}}{\arg \min} \ \|\mathcal{V}(\hat{t}) - \hat{t}\|_2. \tag{3}$$

$\mathcal{S}$ denotes the dictionary of the pre-trained model. The projected prompt is used to construct the instruction for LLM to infer better prompt candidates in the unconstrained semantic vocabulary space.

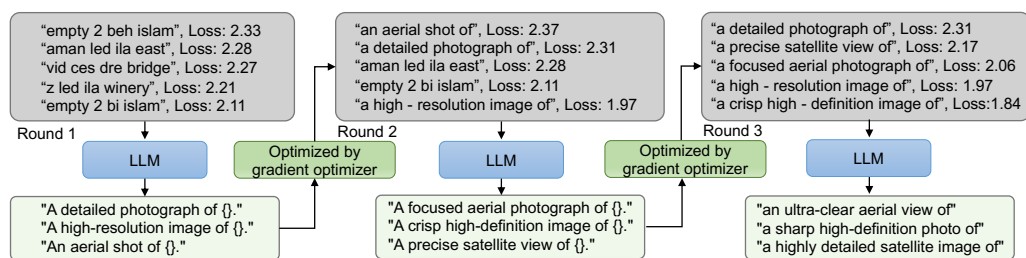

Figure 4: Interpretation of prompts optimized by LLM on EuroSAT dataset.

**Integration with gradient-based optimizer.** Gradient-based optimizers conduct rigorous local-optimal updates on the parameters based on back-propagated gradient. MaaO infers promising candidate prompts by analyzing and generating semantic-related prompts based on currently found solutions. We propose a cooperation workflow of the gradient-based optimizer and MaaO in Algorithm 1.

We connect the two optimizers in two ways. First, the gradient optimizer provides the LLM with the intermediate results in the prompt optimization process, from which LLMs infer more promising candidate prompts. The generated prompts by LLMs assemble task-related semantic contents and provide opportunities to break free from local optimal that may encountered in gradient-based optimization. Second, we restart the gradient-based optimization by using the prompts generated by the LLM optimizer as new initial values of the gradient optimizer to obtain refined prompts based on the LLM-generated ones. Optimizing the prompt based on the two optimizers alternately guides the LLM to progressively exploit better prompts in a more promising area of the search space near the previously found good solutions. The gradient optimizer provides stable convergence for the final learned prompts. Note that the overhead brought by our optimization algorithm compared to the original gradient-based prompt tuning is only about dozens (at most 30) of iterations using the combined optimizer.

## 4 EXPERIMENTS

### 4.1 EXPERIMENTAL SETUP

**Implementation details and baselines.** To comprehensively evaluate the effectiveness of our method, we test close-sourced LLMs GPT-3.5, GPT-4 (OpenAI et al., 2024), and open-sourced Llama2 (Touvron et al., 2023b) as the optimizer LLMs. We employ MaaO in P-tuning (Liu et al., 2023b), Lester et al. (2021), which are the pioneering work of prompt tuning for pre-trained language models. We also apply the optimization methods to prompt tuning methods for the vision-language model, CoOp (Zhou et al., 2022b), TCP (Yao et al., 2024). CoOp is the founder of prompt tuning for vision-language models, and TCP represents one of the state-of-the-art advancements in this realm. Both methods exemplify the use of textual prompting techniques for enhancing vision-language models. For a fair comparison, we fix the original hyperparameter of previous methods, such as pre-trained backbone and prompt module design, and only apply our method as a new optimization strategy. For the configuration of Algorithm 1, the number of rounds $N$ is set as 3, and the iteration for the gradient optimizer $m$ is set as 10. All experimental results are averaged over 3 random seeds. More detailed hyperparameter settings are provided in the appendix.

**Datasets.** For the lauguage model, we conduct experiments over the commonly-used pre-trained model RoBERTa (Liu et al., 2019) on NLU tasks from SuperGLUE (Wang et al., 2020) to test our methods. We apply our prompt optimization algorithm to vision-language pre-trained CLIP (Radford et al., 2021) for adaptation of image classification tasks. We adopt commonly used 10 datasets to comprehensively evaluate our method, including Caltech101 (Li et al., 2004), OxfordPets (Parkhi et al., 2012), StanfordCars (Krause et al., 2013), Flowers102 (Nilsback & Zisserman, 2008), Food101 (Bossard et al., 2014), FGVCAircraft (Maji et al., 2013), SUN397 (Xiao et al., 2010), UCF101 (Soomro et al., 2012), DTD (Cimpoi et al., 2014), and EuroSAT (Helber et al., 2019). Labeled few-shot samples from each class are used as training data for each dataset.

Table 1: Results of prompt tuning pre-trained language model RoBERTa-Large on SuperGLUE dev-set. (PT: P-tuning & Lester et al. (2021)).

| Methods | COPA | BoolQ | RTE | WiC | WSC | Avg. |
|---|---|---|---|---|---|---|
| PT | 61.67 | 62.29 | 55.72 | 53.81 | **64.10** | 59.52 |
| Ours | **68.67** | **63.09** | **58.00** | **55.85** | 63.46 | **61.81** |

Table 2: Result of few-shot prompt tuning vision-language model CLIP on downstream datasets. Top optimization results of different optimizer LLMs are marked with different colors.

| Datasets | Settings | ResNet50 | | | | | ResNet50 | | ViT-B/16 | |
|---|---|---|---|---|---|---|---|---|---|---|
| | | CoOp | Liu et al.(2023a) | Ours(GPT3.5) | Ours(GPT4) | Ours(Llama) | TCP | Ours(GPT3.5) | TCP | Ours(GPT3.5) |
| Eurosat | 1-shot | $50.58_{2.74}$ | 49.0 | $56.27_{2.06}$ | $56.74_{2.28}$ | $55.38_{1.91}$ | $62.79_{2.10}$ | $63.07_{0.56}$ | $65.04_{0.99}$ | $63.06_{1.05}$ |
| | 4-shot | $69.65_{0.73}$ | - | $71.17_{0.84}$ | $72.55_{0.87}$ | $73.39_{0.89}$ | $73.20_{1.40}$ | $74.10_{0.39}$ | $72.42_{0.50}$ | $77.35_{0.26}$ |
| | 8-shot | $72.74_{0.96}$ | - | $74.33_{1.90}$ | $75.99_{1.05}$ | $76.79_{0.67}$ | $77.37_{0.44}$ | $77.95_{0.15}$ | $77.71_{0.02}$ | $79.49_{0.19}$ |
| | 16-shot | $83.57_{0.46}$ | 51.4 | $83.77_{1.18}$ | $85.07_{0.55}$ | $83.95_{0.58}$ | $82.37_{0.29}$ | $83.01_{0.41}$ | $84.43_{0.09}$ | $86.18_{0.15}$ |
| | Avg. | 69.14 | - | 71.39 | 72.59 | 72.38 | 73.93 | 74.53 | 74.90 | 76.52 |
| DTD | 1-shot | $43.13_{1.86}$ | 44.8 | $47.24_{0.37}$ | $44.78_{1.29}$ | $42.47_{1.09}$ | $48.25_{0.32}$ | $48.64_{0.25}$ | $55.06_{1.20}$ | $55.14_{0.35}$ |
| | 4-shot | $53.45_{0.47}$ | - | $54.87_{0.90}$ | $55.04_{0.71}$ | $54.14_{0.21}$ | $60.28_{0.24}$ | $60.30_{0.27}$ | $61.88_{0.05}$ | $63.36_{0.44}$ |
| | 8-shot | $59.38_{1.06}$ | - | $60.30_{0.76}$ | $60.18_{0.29}$ | $60.48_{0.61}$ | $64.38_{0.53}$ | $65.26_{0.20}$ | $68.62_{0.46}$ | $68.56_{0.27}$ |
| | 16-shot | $63.87_{0.24}$ | 44.9 | $64.40_{0.23}$ | $64.30_{0.88}$ | $64.48_{0.73}$ | $68.18_{0.61}$ | $68.45_{0.26}$ | $73.48_{0.14}$ | $73.66_{0.12}$ |
| | Avg. | 54.96 | - | 56.70 | 56.08 | 55.39 | 60.27 | 60.66 | 64.76 | 65.18 |
| Caltech101 | 1-shot | $87.76_{0.92}$ | 89.1 | $87.02_{0.56}$ | $86.87_{0.58}$ | $87.86_{0.39}$ | $89.16_{0.43}$ | $89.58_{0.15}$ | $94.08_{0.23}$ | $94.00_{0.18}$ |
| | 4-shot | $89.05_{0.55}$ | - | $88.72_{0.27}$ | $88.68_{0.24}$ | $89.03_{0.16}$ | $91.15_{0.05}$ | $91.40_{0.23}$ | $95.18_{0.02}$ | $95.38_{0.17}$ |
| | 8-shot | $90.58_{0.52}$ | - | $90.26_{0.59}$ | $90.66_{0.41}$ | $90.25_{0.94}$ | $92.05_{0.26}$ | $91.99_{0.20}$ | $95.39_{0.19}$ | $95.33_{0.07}$ |
| | 16-shot | $91.66_{0.22}$ | 89.5 | $92.28_{0.50}$ | $91.83_{0.05}$ | $92.33_{0.06}$ | $93.25_{0.20}$ | $93.05_{0.02}$ | $95.89_{0.16}$ | $95.71_{0.15}$ |
| | Avg. | 89.76 | - | 89.57 | 89.51 | 89.87 | 91.40 | 91.51 | 95.14 | 95.11 |
| Oxford Flowers | 1-shot | $69.09_{1.57}$ | 67.2 | $71.62_{1.01}$ | $71.28_{1.03}$ | $72.31_{1.22}$ | $78.51_{0.37}$ | $77.94_{0.69}$ | $85.80_{0.56}$ | $87.05_{0.36}$ |
| | 4-shot | $87.00_{0.91}$ | - | $89.38_{0.75}$ | $88.51_{0.43}$ | $88.93_{0.52}$ | $90.85_{0.13}$ | $91.03_{0.20}$ | $94.72_{0.20}$ | $95.09_{0.28}$ |
| | 8-shot | $90.19_{0.34}$ | - | $91.15_{0.63}$ | $90.77_{0.45}$ | $90.41_{0.39}$ | $93.31_{0.05}$ | $93.74_{0.18}$ | $96.14_{0.14}$ | $96.31_{0.04}$ |
| | 16-shot | $93.88_{0.13}$ | 67.4 | $94.42_{0.28}$ | $94.02_{0.31}$ | $94.49_{0.41}$ | $95.38_{0.21}$ | $95.33_{0.15}$ | $97.47_{0.05}$ | $97.54_{0.10}$ |
| | Avg. | 85.04 | - | 86.64 | 86.15 | 86.54 | 89.51 | 89.51 | 93.53 | 94.00 |
| Fgvc Aircraft | 1-shot | $18.38_{0.84}$ | 18.1 | $18.69_{0.67}$ | $18.82_{0.48}$ | $18.50_{0.38}$ | $20.01_{0.28}$ | $20.69_{0.14}$ | $28.90_{0.31}$ | $28.33_{0.21}$ |
| | 4-shot | $21.90_{0.57}$ | - | $22.73_{1.07}$ | $22.77_{0.34}$ | $23.19_{1.03}$ | $25.18_{0.16}$ | $25.72_{0.11}$ | $35.61_{0.60}$ | $35.61_{0.31}$ |
| | 8-shot | $25.15_{0.55}$ | - | $26.53_{0.16}$ | $26.42_{0.54}$ | $27.35_{0.56}$ | $29.97_{0.24}$ | $30.31_{0.18}$ | $39.76_{0.49}$ | $40.69_{0.32}$ |
| | 16-shot | $28.86_{0.59}$ | 18.1 | $31.27_{0.12}$ | $31.24_{0.56}$ | $31.44_{0.78}$ | $34.03_{0.69}$ | $34.36_{0.14}$ | $43.29_{0.22}$ | $43.79_{0.29}$ |
| | Avg. | 23.57 | - | 24.81 | 24.81 | 25.12 | 27.30 | 27.77 | 36.89 | 37.11 |
| Food101 | 1-shot | $72.60_{0.75}$ | 78.3 | $73.86_{0.40}$ | $73.98_{0.53}$ | $72.47_{0.70}$ | $75.90_{0.20}$ | $75.48_{0.13}$ | $85.74_{0.12}$ | $85.40_{0.16}$ |
| | 4-shot | $70.93_{0.41}$ | - | $70.44_{0.36}$ | $70.25_{0.55}$ | $69.76_{40}$ | $76.09_{0.13}$ | $75.99_{0.21}$ | $86.43_{0.15}$ | $86.01_{0.08}$ |
| | 8-shot | $73.97_{0.51}$ | - | $73.12_{0.13}$ | $73.97_{0.30}$ | $72.51_{0.11}$ | $77.34_{0.12}$ | $77.11_{0.17}$ | $86.83_{0.01}$ | $86.67_{0.11}$ |
| | 16-shot | $75.72_{0.15}$ | 78.3 | $75.20_{0.26}$ | $74.94_{0.19}$ | $73.85_{0.27}$ | $78.47_{0.09}$ | $78.44_{0.02}$ | $87.25_{0.15}$ | $87.26_{0.06}$ |
| | Avg. | 73.31 | - | 73.16 | 73.29 | 72.15 | 76.95 | 76.76 | 86.56 | 86.34 |
| Stanford Cars | 1-shot | $55.70_{0.56}$ | 56.2 | $54.74_{0.81}$ | $54.89_{0.93}$ | $54.78_{0.62}$ | $56.37_{0.28}$ | $55.59_{0.27}$ | $68.87_{0.79}$ | $68.05_{0.67}$ |
| | 4-shot | $61.22_{0.54}$ | - | $61.93_{0.18}$ | $61.76_{0.17}$ | $62.01_{0.42}$ | $66.02_{0.26}$ | $66.87_{0.16}$ | $75.25_{0.26}$ | $76.17_{0.20}$ |
| | 8-shot | $65.14_{0.54}$ | - | $65.89_{0.42}$ | $66.82_{0.59}$ | $67.37_{0.60}$ | $71.02_{0.30}$ | $70.80_{0.37}$ | $79.27_{0.29}$ | $79.29_{0.21}$ |
| | 16-shot | $67.97_{0.36}$ | 56.8 | $68.84_{1.10}$ | $69.34_{0.38}$ | $73.30_{0.28}$ | $75.39_{0.20}$ | $75.81_{0.13}$ | $83.79_{0.11}$ | $83.98_{0.23}$ |
| | Avg. | 62.51 | - | 62.85 | 63.20 | 64.37 | 67.20 | 67.27 | 76.80 | 76.87 |
| Oxford Pets | 1-shot | $85.14_{0.78}$ | 88.1 | $85.70_{0.93}$ | $84.38_{0.36}$ | $84.78_{0.53}$ | $86.76_{0.31}$ | $86.16_{0.11}$ | $91.26_{0.40}$ | $90.75_{0.24}$ |
| | 4-shot | $85.37_{0.44}$ | - | $85.03_{0.86}$ | $85.33_{0.88}$ | $84.47_{0.67}$ | $88.29_{0.17}$ | $87.47_{0.03}$ | $92.67_{0.25}$ | $92.65_{0.11}$ |
| | 8-shot | $85.70_{0.53}$ | - | $84.65_{0.20}$ | $84.82_{0.39}$ | $84.58_{0.21}$ | $87.85_{0.10}$ | $87.44_{0.13}$ | $92.91_{0.27}$ | $92.55_{0.13}$ |
| | 16-shot | $86.84_{0.08}$ | 88.3 | $86.37_{0.07}$ | $86.00_{0.32}$ | $85.02_{0.24}$ | $89.63_{0.26}$ | $89.23_{0.13}$ | $93.34_{0.07}$ | $93.19_{0.18}$ |
| | Avg. | 85.76 | - | 85.44 | 85.13 | 84.71 | 88.13 | 87.58 | 92.55 | 92.29 |
| UCF101 | 1-shot | $62.60_{0.88}$ | 60.2 | $61.85_{0.69}$ | $62.12_{0.09}$ | $62.34_{0.45}$ | $64.72_{0.74}$ | $64.51_{0.44}$ | $73.44_{0.54}$ | $72.69_{0.11}$ |
| | 4-shot | $68.75_{0.38}$ | - | $68.25_{0.56}$ | $69.18_{0.79}$ | $68.27_{0.84}$ | $72.57_{0.51}$ | $73.85_{0.11}$ | $80.93_{0.08}$ | $80.77_{0.09}$ |
| | 8-shot | $72.26_{0.30}$ | - | $72.69_{0.19}$ | $72.26_{0.44}$ | $72.58_{0.41}$ | $76.68_{0.01}$ | $77.61_{0.19}$ | $83.18_{0.20}$ | $83.40_{0.10}$ |
| | 16-shot | $74.91_{0.33}$ | 60.5 | $74.82_{0.87}$ | $75.96_{0.14}$ | $74.95_{0.33}$ | $78.91_{0.20}$ | $79.95_{0.01}$ | $85.25_{0.25}$ | $85.13_{0.27}$ |
| | Avg. | 69.63 | - | 69.40 | 69.88 | 69.54 | 73.22 | 73.98 | 80.70 | 80.50 |
| SUN397 | 1-shot | $58.33_{0.76}$ | 61.0 | $58.13_{0.65}$ | $57.25_{0.43}$ | $57.47_{0.79}$ | $60.94_{0.17}$ | $61.13_{0.22}$ | $69.20_{0.19}$ | $69.13_{0.12}$ |
| | 4-shot | $64.48_{0.25}$ | - | $65.21_{0.37}$ | $64.06_{0.40}$ | $64.40_{0.39}$ | $67.12_{0.04}$ | $67.37_{0.11}$ | $73.78_{0.04}$ | $73.94_{0.10}$ |
| | 8-shot | $66.79_{0.23}$ | - | $67.20_{0.24}$ | $66.90_{0.22}$ | $66.79_{0.33}$ | $69.83_{0.12}$ | $69.74_{0.09}$ | $75.78_{0.01}$ | $75.99_{0.04}$ |
| | 16-shot | $68.79_{0.26}$ | 60.8 | $68.39_{0.09}$ | $68.42_{0.18}$ | $68.42_{0.21}$ | $71.97_{0.18}$ | $72.26_{0.15}$ | $76.81_{0.12}$ | $76.69_{0.07}$ |
| | Avg. | 64.60 | - | 64.73 | 64.16 | 64.27 | 67.47 | 67.63 | 73.89 | 73.94 |

## 4.2 MAIN RESULTS

**Prompt optimization for language models.** We employ the proposed optimization method for the prompt tuning of pre-trained language model RoBERTa-large (Liu et al., 2019) and evaluate on the widely used SuperGLUE (Wang et al., 2020) NLU tasks. Previous prompt tuning methods, P-tuning (Liu et al., 2023b) and Lester et al. (2021), use only backpropagated gradient to optimize the prompt. From Table 1, our combined optimization method with GPT-4 as optimizer surpasses vanilla gradient-based optimization on four out of five tasks from SuperGLUE.

Table 3: More results demonstrating the relation of the two collaborative optimizers.

| Methods | EuroSAT | DTD | Oxford_Flowers |
|---|---|---|---|
| Single-start Gradient Optimization | 69.14 | 54.96 | 85.04 |
| Multi-start Gradient Optimization | 70.64 | 55.08 | 85.45 |
| Multi-start Gradient Optimization with Perturbations | 70.33 | 55.42 | 85.71 |
| LLM-based Optimization | 49.21 | 44.16 | 67.05 |
| Ours | 71.39 | 56.70 | 86.64 |

**Prompt optimization for vision-language pre-trained models.** We also compare with prompt tuning methods for vision-language models, (Zhou et al., 2022b; Yao et al., 2024). From Table 2, the results on the "RN50" backbone show that our integrated optimization outperforms existing gradient-based prompt tuning methods at six out of ten benchmark datasets, and the other tasks remain close to the baseline performance. Both close-sourced GPT models and open-sourced Llama2 achieve consistent improvements, demonstrating the effectiveness of our combined optimization framework. TCP (Yao et al., 2024) is one of the state-of-the-art prompt tuning approaches with a stronger backbone. Although the absolute improvement inevitably decreases, our method still brings stable improvements on six out of ten datasets.

We also compare with methods that optimize by LLM only, e.g., Liu et al. (2023a). We list the results of the 1-shot and 4-shot settings reported in their paper since the code has not been released yet, and our reproduced results can not match those published in the paper. Although Liu et al. (2023a) achieves a completely program-free prompt learning method by LLM, their performance in the 16-shot setting is poor. This method merely relies on the inherent deductive ability of LLM, which can not make good use of information in more training samples. In our method, the gradient optimizer can promise a stable convergence by learning from more data. And instructed LLMs can exploit in a more promising sub-region of the solution space according to the intermediate results of the gradient optimizer. Thus, better performance is achieved by the collaborative optimization process.

In summary, the results indicate that our proposed combined optimization approach, which leverages both the local precision of a gradient-based optimizer and the flexible semantics exploration of an LLM-based optimizer, is better than both single methods and outperforms each method individually.

**Interpretation of prompts optimized by LLM.** To further analyze the contribution of LLM-based optimizer in prompt optimization, we list the prompts contained in the instruction and generated by LLM in Figure 4. In round 1, the gradient-based optimizer tend to navigate around senseless prompt tokens, e.g., "beh", "ila", etc. This phenomenon is alleviated after leveraging LLM to infer more meaningful prompts. In round 3, prompts with both interpretability and low loss values are obtained by the collaboration of gradient-based and LLM-based optimizers.

### 4.3 ABLATION STUDY

Our method's sensitivity to the choice of LLMs, prompt tuning baseline methods, and amount of training samples are already shown in Table 2. We provide more ablation results in this section. The ablation experiments are based on the CoOp baseline, using GPT-3.5 as an optimizer. More ablation studies can be found in the Appendix.

**More analysis of the two optimizers.**

To represent the relation of the two optimizers more clearly, we provide the results in Table 3. "Single-start Gradient Optimization" refers to the basic training procedure of prompt tuning. The parameters are initialized and then trained to convergence in a single training run. To eliminate the influence of training protocol and randomness, we extend the single-start training to "Multi-start Gradient Optimization" by incorporating multiple rounds of optimization. The first round trains from initialized parameters, and the successive rounds restart training from retained parameters of the previous round. "Multi-start Gradient Optimization with Perturbations" means we add random noise values sampled from $0.01 * \mathcal{N}(0, 1)$ to the prompt parameters before each restart round for the opportunity to escape from local optima.

From the results, gradient-based learning itself can not significantly benefit from longer training process and random parameter perturbations. The performance gain of our method lies in the collaboration of gradient-based optimizer and the high-level guidance of LLMs.

**Design of instruction.** The instruction from which LLMs infer for new candidates influences the results. We empirically analyze the effect of each component in our instructions in Table 4. Task definition (TD) denotes raw instruction defining the task information. Manual prompt (MP) means LLMs are instructed

Table 4: Ablation on the design of instruction.

| TD | MP | OT | EuroSAT | DTD | Oxford_Flowers |
|----|----|----|---------|-------|----------------|
| ✗ | ✗ | ✗ | 70.33 | 55.42 | 85.71 |
| ✓ | ✓ | ✗ | 70.56 | 55.95 | 86.44 |
| ✓ | ✗ | ✓ | 71.26 | 56.98 | 86.25 |
| ✓ | ✓ | ✓ | 71.39 | 56.70 | 86.64 |

with hand-crafted prompt templates. Optimization trajectory (OT) denotes the intermediate results from the gradient optimizer provided. The results on the first line of Table 4 correspond to no LLM-based optimization, serving as a baseline. The ablation results show that hand-crafted templates, providing prior knowledge of the prompt, and optimization trajectory, providing a timely semantic landscape of currently optimizing prompts, are both important components for ideal performance.

**Rounds of alternating optimization.** We analyze the effect of the alternating rounds $N$ of the two optimizers on the result. Table 5 indicates that the optimal round for each task varies. But more rounds involve more interactions with LLM, providing more candidates prompts. The average performance improves with more rounds generally. We choose 3 rounds as a proper value.

Table 5: Ablation on the rounds of alternating optimization.

| $N$ | EuroSAT | DTD | Oxford_Flowers |
|-----|---------|-------|----------------|
| 1 | 72.51 | 56.33 | 84.45 |
| 2 | 71.69 | 56.13 | 84.82 |
| 3 | 71.39 | 56.70 | 86.64 |
| 4 | 71.50 | 56.41 | 86.89 |

**The timing of interaction between two optimizers.** We explored how the timing of interactions between the LLM optimizer and the gradient optimizer affects optimization results, maintaining a constant total number of gradient optimization iterations (i.e., keeping $m \times N + M$ iterations constant). In reference to Table 6, smaller values of $m$ indicate that the LLM optimizer is involved

Table 6: Ablation on the iterations of gradient-based optimizer.

| $m$ | EuroSAT | DTD | Oxford_Flowers |
|-----|---------|-------|----------------|
| 10 | 71.39 | 56.70 | 86.64 |
| $10^2$ | 69.46 | 56.10 | 83.92 |
| $10^3$ | 64.31 | 54.81 | 82.31 |

early in the optimization process, whereas larger values of $m$ indicate that the LLM optimizer is introduced during the latter stages of the optimization process. We find that larger $m$ may result in candidate prompts in the gradient trajectory with less semantic diversity, which is less effective for proposing LLM to generate more promising candidate prompts. Furthermore, larger $m$ means smaller $M$ for the last round of gradient optimization, which may lead to insufficient convergence of the algorithm, degrades performance. Thus, we employ a smaller number of training iterations to enable the LLM optimizer to offer a rich variety of candidate prompts during the initial stages of optimization.

## 5 CONCLUSION

This paper proposes a collaborative optimization method combining the conventional gradient-based optimizer and inferential LLM-based optimizer. By alternating between the gradient-based and LLM-based optimization process, we combine the local carefulness of gradient- based optimizer and diverse semantic exploration of LLM-based optimizer. LLM-based optimizer mitigates the inherent limitations of gradient-based optimization, such as entrapment in local optima, by inferring high-level guidance from task descriptions and real-time optimization trajectories. We validated our combined optimization method through prompt tuning tasks, where the synergy between LLM-based optimizer and gradient-based optimizer has consistently demonstrated improved performance over competitive baselines. These results underscore the complementary effect of LLM-based optimizer and conventional gradient-based optimization. Our contributions inspire further exploration of the advantages of LLM-based optimization over existing algorithms, paving the way for more effective integration of LLM-based inference into conventional optimization workflows.

**Limitations.** Our proposed optimization method can not be directly employed for adapter-based or LoRA-based fine-tuning methods. A feasible solution for handling higher dimensional parameters in LLM-based optimization needs to be designed. We leave the application of the proposed optimization framework to broader range of optimization problems (*e.g.*, adapters, LoRA) and algorithms (*e.g.*, reinforce learning) as future work.

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

# A APPENDIX

## A.1 MORE EXPERIMENTAL DETAILS

**Instructions used to query LLMs.** The instruction used to query GPT-3.5 and GPT-4 has been shown in Figure 3 of the main text. The instruction for Llama2-7B-chat is provided in Figure 5.

The design of instruction for Llama2-7B is different from GPT-3.5 and GPT-4 since we notice that the instruction following ability of Llama2-7B is weaker. It is more likely to produce unexpected output. Even though we emphasized the desired way of responding to our query, the responses from Llama2-7B still need proper post-processing to obtain the clean returned prompts.

---

**System:** You are a helpful, respectful, and honest assistant capable of proposing new prompts for users.
**User:** Propose new prompts for user. Reply with only the proposed short template, do not reply the loss and accuracy. Keep every template under 8 words. Generate 3 templates that potentially have better image recognition performance. I have a list of text templates with their corresponding loss values and accuracy. They are used for image classification with CLIP model. The templates are arranged in descending order based on their loss value on training samples, where lower loss indicates better quality.

(Insert optimized prompts as optimization trajectories here.)

---

Figure 5: The instruction used to query Llama2-7B-chat in an iteration of optimizing the prompt using LLM.

**Detailed hyperparameter settings.** The backbone models used by CoOp and TCP are ResNet50 and ViT-B/16, respectively. The prompt length is set as 4 for both CoOp and TCP. The training hyperparameters, such as epochs and learning rate, remained the same as the original methods. The number of training iterations $M$ for Algorithm 1 equals the training iterations of the original methods. We set the number of rounds $N$ as 3, and the iteration for the gradient optimizer $m$ is set as 10 for CoOp and 30 for TCP. The prompt length for NLU tasks is set as 8. The experiments are conducted on a V100 GPU. The specific versions of the API we are utilizing are "gpt-3.5-turbo-1106" for "GPT-3.5" and "gpt-4-1106-preview" for "GPT-4".

## A.2 MORE ABLATION STUDY

**Distance function used for token space projection.** The token space projection operator in Eqn. 3 uses L2 distance to find the nearest discrete tokens for continuous prompt embeddings. We also tried to use cosine similarity as a distance function. The results are provided in Table A.2

Table 7: Ablation on distance function used for token space projection.

| Distance Function | EuroSAT | DTD | Oxford_Flowers |
|:---:|:---:|:---:|:---:|
| L2 | 71.39 | 56.70 | 86.64 |
| Cosine | 71.36 | 56.19 | 87.09 |

**Length of the prompt.** We use a default prompt length of 4 for our experiments. We provide the result of our method with a longer prompt in Table A.2.

Table 8: Ablation on the length of the prompt.

| Methods | EuroSAT | DTD | Oxford_Flowers |
|---|---|---|---|
| Gradient-based Search (length 4) | 69.14 | 54.96 | 85.04 |
| Ours (length 4) | 71.39 | 56.70 | 86.64 |
| Gradient-based Search (length 8) | 69.36 | 55.10 | 85.48 |
| Ours (length 8) | 70.52 | 56.38 | 86.90 |
| Gradient-based Search (length 16) | 70.55 | 54.93 | 85.01 |
| Ours (length 16) | 71.54 | 56.24 | 86.45 |

