# OpenReview forum: "LLM as a Complementary Optimizer to Gradient Descent: A Case Study in Prompt Tuning"
_ICLR.cc/2025/Conference — Submitted to ICLR 2025_

### Official Review · Reviewer_Uh1W · 2024-10-29

**Soundness:** 3
**Presentation:** 3
**Contribution:** 2
**Rating:** 5
**Confidence:** 4

**Summary:**

This paper proposes a method to optimize the answers provided by large language models by combining the properties of prompt tuning using an LLM and gradient based optimization. They compare their method against previous works that use either of these methods individually and showcase superior performance to the individual algorithms used in isolation.

**Strengths:**

The paper is well written and presents the ideas in a clear manner.

The idea itself is well motivated and should be easily replicatable.

One potentially really interesting part of this paper is the interpretability mentioned in line 464: This is indeed a unique advantage of the LLM based optimization above what can be provided by EA or Gradient based methods, and could potentially even be interesting to use as a standalone interpretation method (i.e. following the gradient, but keeping the result readable).
We encourage the authors to follow up on this.

**Weaknesses:**

In the abstract you ask the question:
> “ Will this strategy also work well for solving complex non-convex optimization problems?”

This is not sufficiently answered (or arguably answered at all): This paper is not principally about optimization, but rather about prompt tuning. The title is also misleading: This is not a general optimization scheme that happens to have been studied in prompt tuning, but instead a technique _only_ applicable to prompt tuning. I would recommend the authors to remove the references towards more general optimization. Reading the title and abstract of this paper I would expect something closer to https://arxiv.org/pdf/2309.03409 than prompt tuning.

The “Algorithm 1” definition is quite messy: the embedding and token space projection operators are not inverses of each other, so shouldn’t be treated as such. E.g. in line 15 $\tilde p$ is reinitialized despite never being initialized in the first place.


Regarding token-space projection: In line 311 you claim that
> However, it is not feasible to directly provide the soft embedding vectors as input to the LLM,

This might be true with API models, but is not necessarily true for Llama models: As long as the tuned model and the model tuner are of the same type, one could directly input the continuous embeddings. This is done in textual inversion (https://arxiv.org/pdf/2208.01618), and is also the way that the original P-tuning (https://arxiv.org/pdf/2103.10385) worked. Are you also discretizing the P-tuning outputs?
As long as the prompt tuner and the prompt tunee is the same model, I see no reason why the inputs to the tuner have to be discretised first.
For gradient based tuners I would like to see a comparison on the same model with/without discretization since I suspect that the discretization is the true culprit for the performance drops, not the local optimality (indeed recent work, such as https://arxiv.org/pdf/2305.12827 suggests quasi-linearity of models when finetuning around pretrained models).

In line 515ff. you say that:
>A longer training process by gradient-based optimizer may not necessarily benefit the final prompt tuning results. We find that more iterations of gradient-based optimization may result in candidate prompts with less semantic diversity, which is not good for proposing LLM to generate more promising candidate prompts.

Doesn’t this show that the LLM can*not* escape local optima? If the LLM was necessary to escape the local optima, I would expect the LLM to make more of a difference the closer you are to the optimum.

In general, there are not enough comparisons against the state-of-the-art: from the LLM point of view I would expect a comparison against Zhou et al (https://arxiv.org/pdf/2211.01910) which also provide code to reproduce their experiments.
Due to the problem of gradient based optimization in discrete spaces, people have also started moving towards black-box combinatorial solvers, such as Guo et al’s Evolutionary method (https://arxiv.org/pdf/2309.08532, code is also available). Alternatively, you can also look at Deepmind’s PROMPTBREEDER (https://arxiv.org/pdf/2309.16797) which follows a similar idea
The reason I would like an ablation against the e.g. evolutionary method is because as-is it is unclear whether the LLM component of your framework is necessary or whether the advantage simply comes from the optimizer not being gradient based: “Is it necessary to have an LLM, or is it necessary to _not_ use gradients?”.

As a general note: when using APIs such as GPT-3.5 or GPT-4, please also give information of the time the experiments were done. APIs change all the time, so without knowing the timeframe the experiments were conducted in, it becomes hard to evaluate the results 2-3 years later.

In appendix A1 you list the prompt used for Llama2, which is quite different from the GPT-3.5/4 models: you mention that this is due to the inherent differences in the Llama and GPT models, but that opens up another question: How stable are the models to changes in prompts?
The prompt is a quite complex hyperparameter that you do not need in e.g. Evolutionary or gradient based methods, so one should compare different prompts as to not accidentally overfit the used prompt to the explored datasets.

Appendix A2 shows that the performance gap generally reduces as you increase the prompt length: Is there any interpretation for why this might be happening?
If I use 32 tokens, does this close the gap?


How do you set the learning rate for the gradient based systems? I've seen people use the traditional line-search with Wolfe/armijo rules to "get rid" of the learning rate hyperparameter.

**Questions:**

- What happens if you do not discretise the gradient outputs (i.e. follow the original P-tuning protocol)
- How does this method compare to purely combinatorial solvers, like Promptbreeder, Guo et al (https://arxiv.org/pdf/2309.08532), and Zhou et al (https://arxiv.org/pdf/2211.01910)
- How stable is this method against changes in tuning-prompts?
- If I use 32 tokens, does this close the gap?
- How do you set the learning rate for the gradient based systems?

For more details, see the "weaknesses" section

---

> ### Comment · Reviewer_Uh1W · 2024-12-02
>
> Considering the critiques by other reviewers and comments on their reviews I have decided to maintain my original score.

---

### Official Review · Reviewer_LGsA · 2024-10-29

**Soundness:** 3
**Presentation:** 3
**Contribution:** 3
**Rating:** 5
**Confidence:** 4

**Summary:**

This paper proposes an optimization method that combines LLM and gradient descent to address the prompt tuning problem. The LLM generates potential prompts based on the historical path of gradient descent, and then these prompts are used as  restarting point for the next stage of gradient descent. This combined optimization approach ultimately achieves results that match or even surpass the current SOTA method, TCP, on several benchmark tasks.

**Strengths:**

The paper innovatively combines the gradient-based optimizers and the optimization capabilities of LLMs, achieving good results on many benchmarks and providing detailed experimental details.

**Weaknesses:**

The experiments are not very comprehensive. The paper only compares with two methods, CoOp and TCP (I am ignoring the results from Liu et al.), and it only compares with CoOp on Resnet50 and with TCP on ViT-B/16, lacking a comparison with the other method. Additionally, there is no comparison with other methods such as PromptSRC, MaPLe, and KgCoOp.

**Questions:**

- In Table 2, the comparison is only made with the CoOp method on the ResNet50 model and with the TCP method on the ViT-B/16 model. Why was the SOTA method, TCP, not compared on the ResNet50 model?
- Looking at the flow in Figure 4, each time the LLM generates three candidate prompts. When using gradient descent for optimization, is one of these prompts selected, or do all three prompts need to be optimized?
- In Section 3.2, it is mentioned that "the combined method can achieve smaller standard deviations and better absolute performance." From Figure 2, I cannot clearly see the smaller standard deviation. Listing the numbers might be better.
- In Section 4.2, "From Table 1, the results on the 'RN50' backbone show that...," should it be Table 2?
- I don't quite understand the Single-start gradient optimization and Multi-start Gradient Optimization in Table 3. Could you provide a more detailed explanation? Also, in Section 4.3, the sentence "Multi-start Gradient Optimization (w/w.o. Perturbations)." seems incomplete.

I'm willing to improve my score if you address my concerns.

---

### Official Review · Reviewer_PKSg · 2024-10-30

**Soundness:** 3
**Presentation:** 3
**Contribution:** 2
**Rating:** 5
**Confidence:** 3

**Summary:**

This paper presents an innovative approach to non-convex optimization by combining gradient-based methods with large language models (LLMs) to mimic a mentor-doer relationship. The gradient-based optimizer makes locally optimal search, while the LLM suggest possible improved starting points based on instructions that include the gradient-based optimizer's search history. The authors apply this approach to prompt tuning for both text models and vision-language models.

**Strengths:**

1. The paper introduces a novel combination of gradient-based methods and LLM-based planner. This approach is both creative and promising, as it leverages the complementary strengths of gradient-based optimization for local search and LLMs for strategic guidance.

2. The authors perform a comprehensive set of experiments to demonstrate the effectiveness and efficiency of their method. The breadth of experimentation, covering both text model and vision-language model prompt tuning, demonstrate the approach’s versatility and potential applicability across various domains.

**Weaknesses:**

The main limitation of this approach is the unpredictable nature of the LLM-based planner.
1. Based on results in Table 5, additional rounds of optimization do not always yield better outcomes, and N is too small to observe a clear performance trend in the optimization performance.
2. Table 2  shows that smaller models like LLaMA-7B sometimes outperform larger models such as GPT-3.5 and even GPT-4. This suggests that the observed performance gains may not stem from any advanced reasoning ability inherent to the LLMs, but rather from task-specific nuances.
3. Table 6 results show that increasing training iterations of the gradient optimizer leads to worse results, which suggests that the gradient-based optimizer and LLM planner may not be fully compatible, potentially causing interference rather than collaborative improvement.

Overall, I personally believe the paper may overstate the readiness of LLMs as optimization planners, it may be premature to combine LLMs with gradient-based optimizer for complex optimization tasks.

**Questions:**

1. The paper lacks a clear trend in performance improvement with additional optimization rounds.

2. The paper tries to use LLM planner to provide diverse semantic exploration of prompts, how could the author measure the diversity of the prompts generated prompts by the LLM planner?

3. In lines 479-480, there are incomplete sentences that may affect clarity.

---

### Official Review · Reviewer_qQQe · 2024-11-04

**Soundness:** 3
**Presentation:** 3
**Contribution:** 3
**Rating:** 5
**Confidence:** 3

**Summary:**

This paper introduces a collaborative optimization framework that combines gradient descent with LLMs to further enhance prompt tuning. In this method, gradient-based optimizer performs local updates while the LLM provides a new restarting point to help escape local optima. Experiments are conducted on language tasks and vision-language tasks.

**Strengths:**

1.	The proposed method is a hybrid method that combines the strengths of local gradient-based optimization and global LLM-based guidance.

2.	The experiments conducted in the paper are comprehensive.

**Weaknesses:**

1.	Based on the experiment results in both language and vision-language tasks, the improvement of the proposed method is mostly incremental, admittedly with a few cases where the improvement from the proposed method is significant. Additionally, as shown in Table 2, whether the proposed method is better than the baseline CoOp seems to depend on what LLM models are used too.

2.	To better gauge the significance of the improvements, it would be beneficial for the authors to report standard deviations, especially given that some performance gains are as small as 0.07. This would provide a clearer picture of the method's reliability and consistency.

3.	Beyond accuracy metrics, I recommend that the authors also compare the running time and memory usage between the proposed method and baseline methods. This additional analysis would give valuable insights into the efficiency and practicality of the approach in real-world applications.

**Questions:**

1.	One potential baseline could be using random initialization to restart the gradient based optimization method. Have the authors considered that?

2.	Why are some cells in Table 2 for Liu et al. (2023a) missing?


3.	What is "$M$" in the algorithm 1?

---

### Comment · Area_Chair_mJ6N · 2024-12-02
**Discussions between reviewers and authors**

Time for discussions as author feedback is in. I encourage all the reviewers to reply. You should treat the paper that you're reviewing in the same way as you'd like your submission to be treated :)

---

### Meta-Review · Area_Chair_mJ6N · 2024-12-21

**Metareview:**

This paper studies LLM prompt tuning, by combining LLMs and gradient descent methods. Specifically, the authors propose using an LLM to generate the "restarting points" for the gradient descent procedure using the historical trajectory of gradient descent.

Reviewers, while commending the idea being interesting and the presentation being clear, have concerns regarding (1) not enough baseline methods in comparison, and (2) branding of the approach as an "oversell". Author rebuttal addressed some, but not all of the concerns.

After a brief read, I also have doubts in whether the method in its current form can be generalised to solving general non-convex optimisation. This is not because the authors only demonstrated the prompt tuning example, but about how the method is justified empirically.

In my opinion, the major question is about the difference between:

(a) using an LLM to **generate** the "restarting points" using the historical trajectory of gradient descent;

(b) using an LLM to **evaluate** the historical trajectory of gradient descent, and then **select** the "restarting point" with the highest fitness.

The proposed approach conducts the first step of option (b) and then feeds in the fitness results to an LLM for "restarting point" generation. The advantage of option (a) is that the generated "restarting point" can be different from the gradient descent history, so might be helpful in terms of jumping out of local optima.

However, first there's no theoretical guarantee that LLMs can do so. Second, even empirically, I think it is critical to continue the second step of option (b) to truly evaluate the effectiveness of an LLM for generating a better "restarting point" compared to the historical solution with best fitness evaluated by an LLM. Also if the non-convex optimisation task is not about prompt tuning, how can LLM help even in the fitness evaluation (i.e., the first step of option (b)) -- perhaps this part also requires some task-specific template design?

Overall I found the idea very interesting, in general I'm curious to see whether this method really works. I encourage the authors to revise their work by incorporating reviewer suggestions and, answering my questions above.

PS --  please provide the error bar results for your 3 seed averaged results to show that your experimental results are indeed statistically significant.

**Additional Comments On Reviewer Discussion:**

Author rebuttal addressed some, but not all of the concerns. In AC-reviewer discussion period none of the reviewers replied to my questions.

---

### Decision · Program_Chairs · 2025-01-22

Reject